

# Ice mélange melt drives changes in observed water column stratification at a tidewater glacier in Greenland

Nicole Abib[1], David A. Sutherland[1], Rachel Peterson[1], Ginny Catania[2], Jonathan D. Nash[3], Emily L. Shroyer[4], Leigh A. Stearns[5], Timothy C. Bartholomaus[6]

[1]Department of Earth Sciences, University of Oregon, Eugene, 97405, USA
[2]Department of Earth and Planetary Sciences, University of Texas, Austin, 78712, USA
[3]College of Earth, Ocean, and Atmospheric Sciences, Oregon State University, Corvallis, 97331, USA
[4]Office of Naval Research, Arlington, 22217, USA
[5]Department of Geology, University of Kansas, Lawrence, 66045, USA
[6]Department of Earth and Spatial Sciences, University of Idaho, Moscow, 83844, USA

*Correspondence to*: Nicole Abib (nabib@uoregon.edu)

**Abstract.** Glacial fjords often contain ice mélange, a frozen conglomeration of icebergs, brash ice, and sea ice, that have been postulated to influence both glacier dynamics and fjord circulation through coupled mechanical and thermodynamic processes. Ice mélange meltwater can alter stratification of the water column by releasing cool, fresh water across a range of depths in the upper layer of the fjord. This meltwater input can subsequently modify the depth at which the subglacial discharge plume reaches neutral buoyancy and therefore the underlying buoyancy-driven fjord circulation and heat exchange with warm ocean shelf waters. Despite a spate of recent modelling studies exploring these proposed feedbacks, we lack in situ observations quantifying changes to the water column induced by ice mélange meltwater. Here we use a novel dataset collected before and after the melt, breakup, and down-fjord transport of an ephemeral ice mélange in front of Kangilliup Sermia (Rink Isbræ) to directly investigate the extent to which ice mélange meltwater can modify glacier-adjacent water properties. We find that even a short-lived ice mélange (4 days) can cause substantial cooling (0.18 °C) and freshening (0.25 g kg$^{-1}$) of the water column that leads to stratification change down to the depth of the outflowing discharge plume. We compare our observations to an adjacent fjord, Kangerlussuup Sermia, where ice mélange seldom forms in the summertime, and show that the presence or absence of ice mélange melt creates fundamental differences in their upper layer hydrography. These observations provide critical constraints for recent modelling studies that have suggested ice mélange meltwater needs to be included in ocean circulation models for glaciers with deep grounding lines and high ice fluxes, which are precisely the glaciers exhibiting the largest magnitude terminus retreats at present.

## 1 Introduction

Ongoing observations have documented the rapid breakup of Greenland's ice tongues (e.g., Wilson et al., 2017; Millan et al., 2023) and the associated dynamic thinning and retreat of its marine terminating outlet glaciers (King et al., 2020; Greene et al., 2024). The rapid retreat of these glaciers can be attributed to environmental forcings occurring at the ice-ocean boundary (Nick et al., 2009; Murray et al., 2010; Carnahan et al., 2022). One key forcing is proposed to be a reduction in the persistence



of rigid ice mélange, a (semi-)permanent conglomeration of icebergs, brash ice, and sea ice at glacier termini that persists for weeks to years (Joughin et al., 2008; Amundson et al., 2010). Although several studies have suggested that meltwater from
icebergs can alter the ocean forcing near tidewater glaciers (Davison et al., 2020, 2022; Kajanto et al., 2023; Hager et al., 2023), the influence of ice mélange meltwater and its temporal variability have been neglected. By volume, ice mélange primarily consists of deep-keeled icebergs, suggesting previous studies estimating subsurface iceberg melt are relevant to understanding how icebergs modify glacier-adjacent fjord waters (e.g., Enderlin et al., 2016; FitzMaurice et al., 2016; Moon et al., 2018; Cenedese and Straneo, 2023). However, many of these studies are either model-dependent or rely on indirect
measurements or parameterizations of iceberg melt rates (Moon et al., 2018; Jackson et al., 2020). Nevertheless, the processes by which a dense and rigid conglomeration of icebergs, such as ice mélange, influence both glacier dynamics through providing physical resistance to glacier flow and ocean stratification by providing a sustained source of cool and fresh water within fjords remains largely unexplored.

Ice mélange can influence the freshwater export from of Greenland's glacial fjords by releasing meltwater over a range of depths in the water column, often below the main pycnocline. Changes in the freshwater flux exiting these glacial fjords can enhance exchange with warm ocean shelf waters by influencing fjord stratification and subsequently the depth and velocity at which the subglacial discharge plume exits the fjord (Straneo et al., 2012; Cowton et al., 2016; Carroll et al., 2017; Slater et al., 2022). This enhanced exchange can in turn influence the behavior of Greenland's outlet glaciers by increasing submarine
melting of glaciers and ice mélange, thereby creating a complex feedback loop between the ocean, glacier, and ice mélange itself (Amundson et al., 2010; Howat et al., 2010; Davison et al., 2020; Wood et al., 2021). At present, potential feedbacks due to ice mélange presence are commonly neglected in ocean-glacier models at the fjord and coarser scales (Carroll et al., 2015, 2017; Krug et al., 2015; Xu et al., 2013; Slater et al., 2017a, b; Kimura et al., 2014; Slater et al., 2016; Bao and Moffat, 2024). However, a recent model study found that including iceberg meltwater increased the net up-fjord heat flux in Sermilik Fjord
by ~10% (Davison et al., 2020). Therefore, understanding the relationship between ice mélange meltwater and buoyancy-driven circulation forced by subglacial discharge has important ramifications for long term glacier stability.

Here we use an opportunistic hydrographic dataset collected before and after the melt, breakup, and down-fjord transport of an ephemeral ice mélange to investigate the extent to which ice mélange meltwater can modify the near-glacier water column.
We show that the ice mélange adds cool and fresh submarine iceberg meltwater that enhances the stratification down to the depth of the outflowing subglacial discharge plume. We then compare these observations to those of an adjacent fjord, where ice mélange seldom forms in the summertime, to show that ice mélange meltwater creates fundamental differences in their upper layer hydrography despite their proximity and shared source of offshore ocean waters. We suggest that in addition to the current scientific focus on long-lived ice mélange events in front of large outlet glaciers like Sermeq Kujalleq and Helheim
Glacier, the changes in stratification of the water column induced by even a four-day ephemeral ice mélange event show that





more work should be done to quantify the frequency, duration, and oceanographic implications of these short-lived events in proglacial fjords.

## 1.1 Physical setting

Kangilliup Sermia (also known as Rink Isbræ: 'RNK') is a deeply grounded (~1,000 m; Morlighem et al., 2017, 2022) and
fast flowing glacier in the Uummannaq District of Central West Greenland (Fig. 1), reaching velocities of up to 4,200 m yr⁻¹ (Bartholomaus et al., 2016) with a corresponding ice flux of 11.6 Gt yr⁻¹ in the summer (Wood et al., 2021). Kangilliup Sermia terminates in Karrats Isfjord, which branches off from Uummannaq Bay ~70 km from the glacier's terminus. The neighboring Kangerlussuup Sermia ('KAS') is a much smaller outlet glacier, with a maximum grounding line depth of 330 m (Morlighem et al., 2017, 2022) and average velocity of 1,800 m yr⁻¹ (Bartholomaus et al., 2016). Several sills are present in the proglacial
fjord of Kangilliup Sermia, the shallowest being at a depth of ~400 m at distance of 50 km from the glacier's terminus (~1,000 m; Morlighem et al., 2017, 2022). Although it is a shallower fjord overall, Kangerlussup Sermia also has a ~400 m deep sill near the fjord's mouth, which is deeper than the grounding line depth of the glacier (Morlighem et al., 2017, 2022). While both fjords share the same offshore ocean conditions, the difference in grounding line depth between the two results in the subglacial discharge plume at Kangilliup Sermia equilibrating at a neutral buoyancy depth of ~120-220 m, while the plume reaches the
surface of the fjord at Kangerlussuup Sermia (Chauché et al., 2014; Carroll et al., 2016; Jackson et al., 2017; Slater et al., 2022).

The two neighbouring fjords exhibit contrasting ice mélange seasonality and characteristics. Kangilliup Sermia typically has ice mélange present from January to June (Fried et al., 2018). After the breakup of winter ice mélange, ephemeral ice mélange
forms episodically after large, full-thickness calving events during the summer months until the reformation of the more persistent winter ice mélange. When ice mélange is present, the icebergs within it have average keel depths of 28-300 m (Sulak et al., 2017). Kangerlussuup Sermia similarly has ice mélange present only between January to June, with no summertime mélange observed (Fried et al., 2018). When present, the icebergs from Kangerlussuup Sermia are much smaller, with average keel depths of 28-190 m (Sulak et al., 2017). One reason for this contrast in ephemeral ice mélange occurrence is likely the
difference in calving style between the two glaciers, with calving at Kangerlussuup Sermia restricted to smaller serac collapse-style events, and calving at Kangilliup Sermia often consists of capsizing slab-style, full thickness events (Fried et al., 2018).



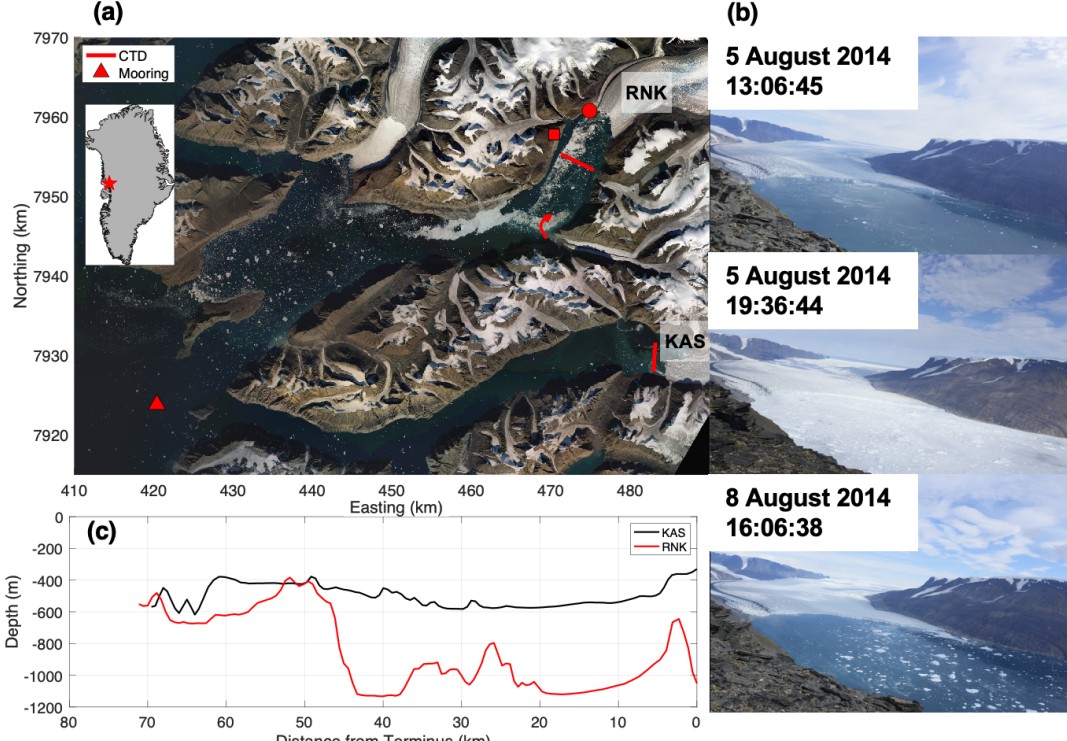

**Figure 1: (a)** True color Landsat 8 image from 07 August 2014, showing the proglacial fjords of Kangilliup Sermia (RNK) and Kangerlussuup Sermia (KAS) overlaid by the location of repeat CTD transects (red line), mooring deployed outside of sill (red triangle), meteorological station (red circle), and time-lapse camera (red square). Red arrow shows direction of riverine sediment plume deflection and inset shows location of image in Greenland. **(b)** Imagery from a time-lapse camera (full video in supplement) shows how the ice mélange forms over the course of 6 hours in front of Kangilliup Sermia following an iceberg calving event on 05 August 2014. **(c)** Thalweg bathymetry of Kangilliup Sermia and Kangerlussuup Sermia extracted from BedMachine v5 (Morlighem et al., 2017, 2022).

## 2 Materials and methods

### 2.1 Ship-based observations

To quantify changes to the water column induced by ice mélange melt, profiles of temperature ($T$) and salinity ($S$) with depth were collected from the proglacial fjords of Kangilliup Sermia and Kangerlussuup Sermia in the summer of 2014 (Fig. 1). A cross-fjord transect was repeated within 5 km of the glacier termini using an RBR XR-620 conductivity-temperature-depth (CTD) sensor (Fig. 1; as described in Bartholomaus et al., 2016; Carroll et al., 2018). The two transects bounding the ephemeral ice mélange event here were collected in front of Kangilliup Sermia, on August 4, 2014, approximately one day before the ice mélange formed, and August 11, 2014, two days after icebergs from the ice mélange cleared the fjord. To characterize the ambient water conditions entering these two fjords, $T$ and $S$ outside of each sill where the fjords branch were recorded with co-located CTD casts and a mooring deployed at a depth of 535 m between September 2013-July 2015 (Fig. 1a). Hydrographic





measurements were converted to potential temperature and absolute salinities following the Thermodynamic Equation of Seawater 2010 (TEOS-10; McDougall and Barker, 2011). Transects captured water column properties from 2 m to a depth of 800 m. To quantify changes in water column stratification, we use the square of the Brunt Vaisala frequency, $N^2$, defined as follows:

$$N^2 = -\frac{g}{\rho_o}\frac{\partial \rho}{\partial z} \tag{1}$$

where $\rho$ is seawater density, $\rho_o$ is the average density of the water column, $g$ is the gravitational constant, and $z$ is depth. This gives us a measure of the stability of a parcel of fluid to vertical displacement and is a measure of how stratified the water column is, with higher frequencies indicating a larger gradient in density with depth.

We use *T-S* diagrams to analyze whether any observed freshening and cooling of the water column (Fig. 2) was due to the input of meltwater, as opposed to subglacial discharge and/or ambient ocean waters (Gade, 1979; Jenkins, 1999; Straneo and Cenedese, 2015). The calculation of change in exact meltwater fraction due to ice mélange melt was precluded, as typical water mass decomposition methods are under-constrained in Greenland's glacial fjords where only two conservative tracers (in this case potential temperature and absolute salinity) are recorded (Beaird et al., 2015). Therefore, we compare only relative changes in meltwater content due to ice mélange melting.

To assess the general circulation pattern in the fjord, ocean velocity data were collected via two downward-looking 300 kHz RDI Workhorse Acoustic Doppler Current Profilers (ADCP). One ADCP was ship-mounted (SADCP) and typically observed velocities ~150 m depth, while the other ADCP was set-up in a lowered ADCP (LADCP) mode connected to the CTD cage. The LADCP collected velocity profiles coincident with each CTD cast, which allowed us to obtain velocity data much deeper in the water column. We combine the two velocity records here by interpolating them onto a grid with 250 m horizontal, 5 m vertical spacing, and rotated into an along and across-fjord coordinate system.

## 2.2 Subglacial plume model

To predict the depth of neutral buoyancy of the upwelling subglacial discharge plume from Kangilliup Sermia, we use a buoyant plume model (Slater et al., 2016) with a line plume geometry (Jackson et al., 2017). We initialize our plume model with a 250 m wide line plume and vary its width between 100-500 m for sensitivity testing (Slater et al., 2022). The initial stratification for the plume model comes from transect-averaged hydrographic profiles from the post-ice mélange CTD casts. Subglacial discharge values for the plume model come from Carroll et al. (2016), who integrated the daily surface runoff from the 1 km downscaled Regional Atmospheric Climate Model (Noël et al., 2015), and assumes that all surface runoff drains immediately to the glacier bed through a probability-based hydrologic catchment. Using these initial inputs, we calculate the neutral buoyancy depth of the plume (where it equilibrates into the fjord) by finding the depth at which the density of the waters in the plume is equal to that of the ambient stratification.



### 2.3 Ice mélange melt rate

To determine whether the change in our hydrographic observations could realistically be explained by ice mélange melt in the proglacial fjord of Kangilliup Sermia, we estimate iceberg melt rates using common parameterizations most recently compiled

by Moon et al. (2018). The parameterized iceberg melt rate results in a depth-varying melt rate that includes wave driven melt, convection driven melt both above and below the waterline, solar radiation driven melt above the waterline, and turbulence driven melt beneath the waterline. As inputs to these parameterizations, we use the average air temperature and wind speed just prior to the ice mélange event taken from a meteorologic station on the north side of the fjord (Fig. 1a). For the initial ocean conditions, we utilize the $T$ and $S$ profiles collected before the ice mélange event and consider the fjord to be 80%

covered by sea ice, icebergs, and bergy bits between the transect and glacier terminus as seen in Landsat 8 imagery from August 5, 2014. In the model higher fjord ice concentrations lead to less melting through wave driven erosion but increase the influence of turbulent subsurface melting due to the relative difference between the water column velocity and iceberg motion, akin to an ice mélange (Moon et al., 2018). We vary the keel depth of the modeled icebergs between 50 m and 400 m to account for variation in iceberg size within the ice mélange. Finally, we average the melt rate over the depth of the iceberg keel to

obtain a depth-averaged melt rate for each iceberg depth class.

Using the depth-averaged iceberg melt rate in combination with an assumption of conservation of volume (Eq. 2) and salt (Eq. 3), we calculate the volume of ice melt ($V_{melt}$; Eq. 4) in the fjord needed to produce the observed salinity changes.

$$V_{ocean_1} + V_{melt} = V_{ocean_2} \tag{2}$$

$$V_{ocean_1}S_1 + V_{ocean_2}S_2 = 0 \tag{3}$$

$$V_{melt} = \left(\frac{\Delta S}{S_1}\right) V_{ocean_1} \tag{4}$$

Here, subscripts 1 and 2 indicate times 1 and 2 of each transect, $V_{ocean}$ is the ocean volume considered, $S$ is the salinity averaged over that volume, and $\Delta S = S_2 - S_1$. We consider the volume of ocean water between the ocean transect and the glacier terminus between a depth of 5-200 m, where the observed $T$ and $S$ changes occurred above the depth of the outflowing

discharge plume. We vary the depth range considered between 100-300 m to test the sensitivity of this control volume on our calculation. We neglect the influence of freshwater runoff from terrestrial and subglacial sources into the control volume, as they do not substantially change over the duration of the ephemeral ice mélange event described here. We then calculate the percentage of the initial calving event volume ($V_{calv}$) that melted ($V_{melt}$):

$$\% \; Ice \; Melted = {V_{melt}}/{V_{calv}} \tag{5}$$


Finally, we calculate the length of time ($t$) needed to melt this volume of ice:

$$t = {V_{melt}}/{(A_{ice} * \dot{m})} \tag{6}$$



where $A_{ice}$ is the area of the fjord covered by ice mélange, and $\dot{m}$ is the estimated iceberg melt rate. The variables $V_{ocean}$, $V_{calv}$, and $A_{ice}$ were determined by manually digitizing the area between the ocean transect and the glacier, the area of the terminus

that changed after the iceberg calving event, and the area of the fjord occupied by ice mélange, respectively using Landsat 8 optical imagery captured on August 5, 2014, and August 7, 2014. These calculations allow us to see if the ephemeral ice mélange event investigated here could have produced the volume of melt necessary to explain the observed freshening of the water column within a duration of time that is similar to the spacing between our hydrographic observations.

## 3 Results

An ephemeral ice mélange formed in the proglacial fjord of Kangilliup Sermia on August 5, 2014, following an iceberg calving event that took place ~2.3 km from the southern edge of the glacier's terminus at ~14:00 GMT. A volume of 3.9 x $10^8$ m$^3$ of ice (assuming full-thickness calving) calved into the fjord and froze into place over the course of 30 minutes (Fig. 1; Supplementary Video). This ice mélange remained frozen in place for ~12 hours, until August 6, 2014, at ~02:00 GMT, at which point it began to break up and move down-fjord. The majority of the ice mélange was transported out-fjord along the

north side wall, but a recirculation in the surface current was seen to transport portions of the ice mélange back towards the terminus along the south side of the fjord. All icebergs from the original calving event were cleared from the fjord after August 9, 2014, and post-event hydrographic observations were collected on August 11, 2014.

Following the formation, breakup, and down-fjord transport of this ephemeral ice mélange event, we observe a freshening and

cooling of the water column. Hydrographic measurements taken after this ice mélange event show that the average water column $T$ and $S$ decreased by 0.19 °C and 0.18 g kg$^{-1}$ respectively (Figs. 2a, b, S1; from 2.35 ± 0.44 to 2.15 ± 0.48 °C and 34.42 ± 0.32 to 34.25 ± 0.48 g kg$^{-1}$ between a depth of 5-800 m). The most significant changes to the water column occurred above a depth of 200 m, where the average $T$ cooled by 0.18°C (from 1.75 ± 0.31 to 1.57 ± 0.31 °C) and $S$ decreased 0.25 g kg$^{-1}$ (from 33.95 ± 0.23 to 33.70 ± 0.50 g kg$^{-1}$). These hydrographic changes coincided with increased stratification to a depth

of 100 m, with the most significant departure from pre-mélange conditions at depths <60 m (Fig. 2c).





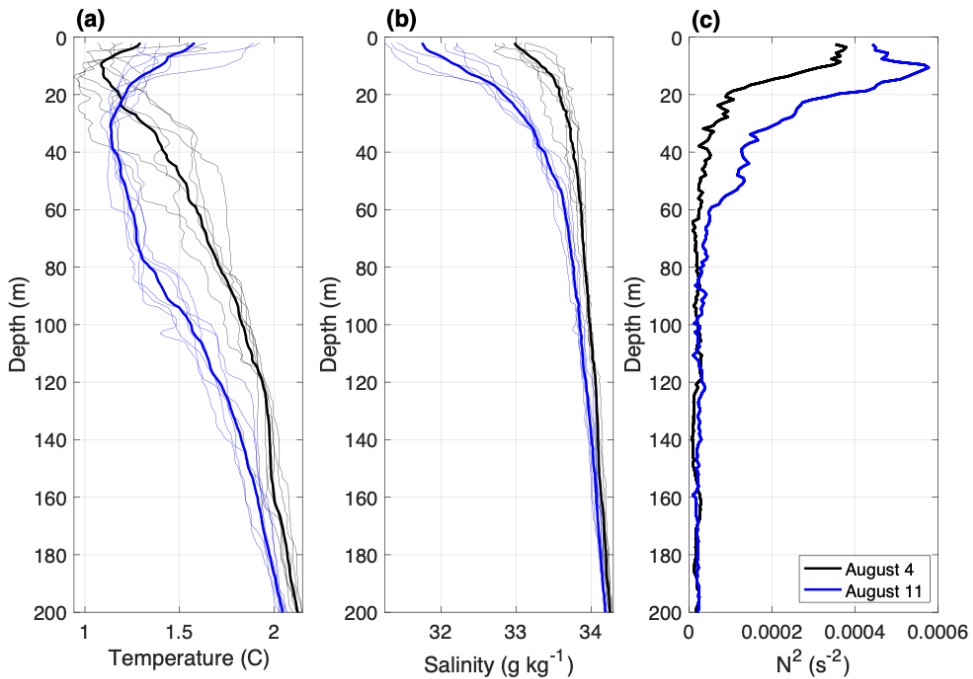

**Figure 2: Changes in $T$ (a), $S$ (b), and $N^2$ (c) pre- (black line; August 4) and post-event (blue line; August 11; full profiles shown in Fig. S1). Thin lines indicate individual CTD casts across the width of the glacial fjord, while thick lines indicate the average. In (c), $N^2$ has been smoothed with a 10 m moving average.**

Below 400 m, there is no significant change between the pre- and post-ice mélange CTD casts (dS = -0.04 g kg⁻¹ and dT = -0.07 °C; Figs. 3a, S1). Between a depth of 280-320 m (Fig. 3a label I), the water properties move towards the runoff mixing line in both the pre- and post-ice mélange event CTD casts, indicating the location where the discharge plume enters the water column horizontally. Above this depth, between 20-280 m (Fig. 3a label II), the water column properties then nearly parallel the meltwater mixing line. In particular, between a depth of 50-100 m, where the largest changes in stratification occur, the

post-event water column properties plot further down the meltwater mixing line (Figs. 3b and S2a), indicating the addition of meltwater in the water column. Starting at a depth of 10 m and 30 m in the pre- and post-ice mélange event CTD casts, respectively, the water column becomes warmer again and shifts back towards the runoff mixing line, suggesting the mixing of the ice mélange melt-modified waters with freshwater input at the fjord surface (Figs. 3a label III, 3b, and S2).



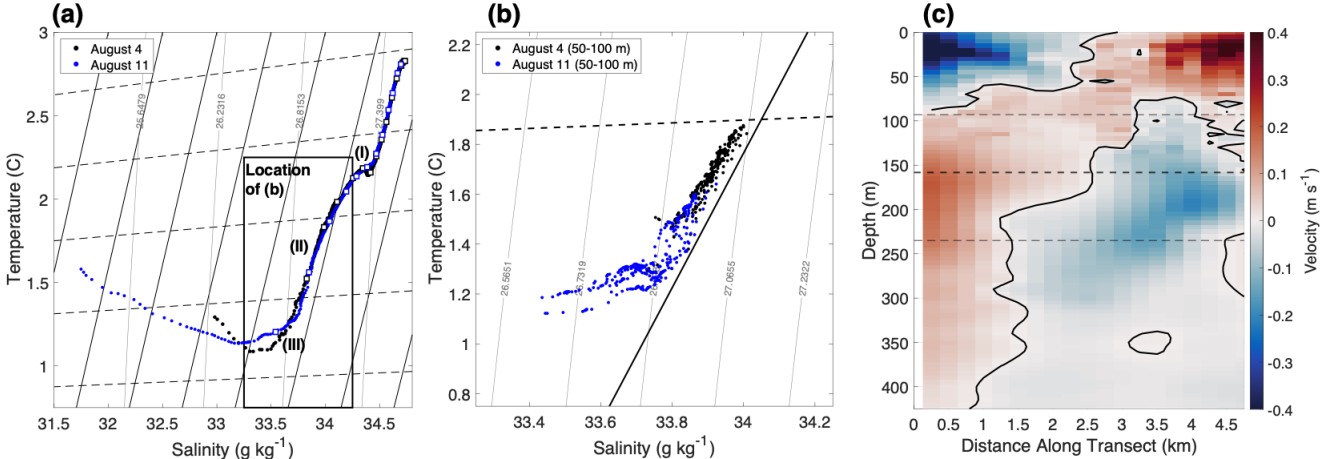

**Figure 3: (a) Average water column properties pre- (black dots) and post-event (blue dots) overlaid on isopycnal contours (density in kg m⁻³ –1000). Mixing lines between freshwater discharge (dashed lines) and submarine melting of ice (solid lines) with ambient ocean water are included, and a white square marker indicates the depth at 50 m intervals between 0-800 m. Roman numerals correspond with locations described in text. (b) Zoom-in on (a) with data only shown between depths of 50-100 m. (c) Gridded ADCP-derived along-fjord velocity on August 11, 2014, with positive values indicating flow toward the glacier and a black contour at 0 m s⁻¹. Distance across fjord increases from southward. The location of the predicted minimum (93 m), maximum (235 m), and mean (158 m) depth of neutral buoyancy of the subglacial discharge plume is shown (black dashed lines).**

A transect of water column velocities from August 11, 2014, across the fjord shows a recirculation gyre present at the surface, with water flowing out-fjord along the northern side at a rate of $0.142 \pm 0.063$ m s⁻¹ and in-fjord on the southern side at $0.154 \pm 0.049$ m s⁻¹ at depths <60 m (Figs. 3c and S3a; additional ADCP transects shown in Fig. S4). The fjord-wide average flow direction is towards the glacier at the surface and out-fjord between a depth of ~160-315 m at rates of $0.013 \pm 0.012$ m s⁻¹ and $0.017 \pm 0.011$ m s⁻¹ respectively. Beneath a depth of 450 m, the average flow direction is towards the glacier at $0.016 \pm 0.009$ m s⁻¹ (Fig. S3a). In the across-fjord transect, we see a signature of the outflowing plume at a depth of 100-325 m, which corroborates the shift towards the runoff mixing line at a depth of ~320 m in *T-S* space (Figs. 3a, c, and S3a). Further, buoyant plume theory suggests that the plume reaches neutral buoyancy depth at 158 m when initialized with the post-ice mélange hydrographic observations, a line plume width of 250 m, and average subglacial discharge of 103 m³ s⁻¹ (Fig. 3c). Sensitivity tests indicate the subglacial discharge plume always reaches neutral buoyancy subsurface (Fig. 3c) over a large range of line plume widths (100-500 m) and subglacial discharge magnitudes (25-450 m³ s⁻¹).

Modeled iceberg melt rates vary between 0.08-1.40 m d⁻¹ (Fig. S3) using the fjord velocity profiles averaged over the northern half, southern half, and entire fjord, as well as the average air temperature and wind speed just prior to the ice mélange event (Fig. S5). Iceberg melt rates are highest at the surface ($0.78 \pm 0.51$ m d⁻¹ above 25 m), reaching a secondary maximum at 200 m depth ($0.29 \pm 0.10$ m d⁻¹). We further evaluate these predicted melt rates for an iceberg with a modeled keel depth of 200 m to investigate the same depth range over which we observe changes to the water column properties following the ephemeral



ice mélange event. For an iceberg with a keel depth of 200 m, we find depth-average modeled iceberg melt rates of 0.18-0.36

m d$^{-1}$, with the highest melt rates coinciding with the velocity profile taken from the southern half of the fjord.

Using the pre- and post-ice mélange salinities ($S_1$ = 33.95 g kg$^{-1}$ and $S_2$ = 33.70 g kg$^{-1}$) as inputs to Eq. (4), we find $V_{melt}$ = 5.47 x 10$^7$ m$^3$. This volume of ice melt is equivalent to 14% of the initial calving event volume that triggered this ephemeral ice mélange (Eq. 5). At the melt rate estimated ($\dot{m}$ = 0.18-0.36 m d$^{-1}$), this volume of ice would have taken 5.1-10.5 days to

melt completely (Eq. 6). Changing the depth range considered for the conservation of salt calculation only has minor effects, shortening the time to 4.3-8.8 days if we only considered changes above 100 m, and increasing it to 5.7-11.7 days for the upper 300 m. While these time spans for complete melting of the ice mélange are longer than the 4-day window that the ice mélange persisted in the proglacial fjord, we note that not all icebergs from the mélange melted completely and many were exported out of the glacial fjord.

**4 Discussion**

We use an opportunistic dataset of near-terminus fjord hydrography observations bookending an ephemeral ice mélange event in the proglacial fjord of Kangilliup Sermia to show that the addition of meltwater from ice mélange over the course of one week causes cooling and freshening of the water column that leads to enhanced stratification. We investigate other mechanisms that could explain these changes, including 1) water mass advection, 2) vertical or horizontal mixing of the water column,

and/or 3) enhanced freshwater input from subglacial discharge or surface runoff. Yet, we demonstrate below that none of these processes explain the observed hydrographic changes. This implies that even short-lived ice mélange events can alter the surface water column stratification, and this can help explain the large-scale differences in water properties between glacial fjords around Greenland and other locations where iceberg concentrations are high (e.g., Carroll et al., 2018). We discuss these implications with a case study of comparing neighbouring fjords with different stratification profiles, Kangilliup Sermia and

Kangerlussuup Sermia, and provide the first observational evidence of the complex feedbacks between ice mélange characteristics, discharge plume-driven circulation, and melt rates at the glacier front.

**4.1 Alternative water column transformation mechanisms do not explain stratification change**

First, we find that neither vertical nor horizontal advection is likely to explain the observed water column transformation. To discern whether the enhanced meltwater presence between a depth of 50-100 m could be attributed to any vertical movement

of the density field (i.e., shoaling or deepening isopycnals), we compared the water column properties at this depth range after the ice mélange event to the water column both above and below this depth prior to the additional meltwater input (Fig. S2a). We find that while the shallow water column properties (20-50 m) share some overlap with the post-mélange properties in *T-S* space, the water column after the ice mélange event is closer to the meltwater mixing line. This suggests that while deepening of the isopycnals from above could have contributed to the changes observed here, enhanced meltwater input from icebergs





also plays a key role in altering the water column properties in this depth range. In addition, we see no evidence that these changes to the water column post-ice mélange event were caused by shoaling of the deeper water column (Fig. S2a). The time series of $T$ recorded at the offshore mooring varies by only ~0.25°C, with no significant changes before and after the ice mélange event (Fig. S6). Furthermore, prior work has shown that Kangilliup Sermia's proglacial fjord is hydraulically controlled in summer months, so that little offshore water is being transported over the sill into the fjord (Carroll et al., 2018).


Second, we see no substantial evidence that local mixing of water masses led to the observed freshening and cooling of the water column. To estimate the impact of horizontal mixing on the water column, we compare the across-fjord $T$ and $S$ gradients to the average temporal change between the two hydrographic transects. The mean temporal change post-event is ~2x larger than any across-fjord variation (Table S1).


Finally, we find that additional freshwater input from subglacial discharge and surface runoff cannot explain the observed cooling and freshening either. A time series of subglacial discharge from Kangilliup Sermia shows a slight peak in magnitude (increase by $17 \pm 4.7\%$) between our two hydrographic surveys (Fig. S7). However, prior work in this fjord has shown that the addition of subglacial discharge results in a warm temperature anomaly, opposite to the cooling signal observed here (Carroll
et al., 2016). Furthermore, both the hydrographic observations and buoyant plume modeling presented here (Figs. 3c and S8) and previously (Carroll et al., 2016; Slater et al., 2022) suggest that the subglacial discharge plume reaches a neutral buoyancy deeper than the observed cooling and freshening signal.

The magnitude of surface runoff (Mankoff et al., 2020) is only 1-2% of the subglacial discharge (Fig. S9). Due to the strength
of the stratification we observe, this magnitude of surface freshwater input is unlikely to cause a cooling of the water column down to the depth observed here (~200 m). It is possible, however, that this surface freshwater could cause the warm, fresh anomaly seen in the upper 20 m of the water column, which is also subject to variable atmospheric heat fluxes (Figs. 2a, b, and S2b). While not responsible for the observed cooling and freshening observed here, the addition of subglacial discharge and surface water to the fjord might play an indirect role by controlling both the net outflow away from the glacier, as well as
any recirculation and residence time of icebergs in the fjord (Fig. 3c; further discussed in Sect. 4.2.1).

### 4.2 Short-term ice mélange events as a driver of water column change

Despite the short-lived nature (4 days) of the ice mélange event, our hydrographic observations show a clear cooling (0.18 °C) and freshening (0.25 g kg$^{-1}$) of the upper 200 m of the water column following the addition of 3.9 x $10^8$ m$^3$ of calved ice into the fjord (Figs. 2a and b). These changes to the water column led to increased stratification down to a depth of 100 m (Fig. 2c)
and $T$-$S$ analysis suggests that this observed freshening and cooling of surface waters can be explained by the submarine melting of icebergs within the ice mélange matrix (Fig. 3b). Prior remote sensing studies in this fjord confirm that typical iceberg keels extend to depths of 28-300 m on average (Sulak et al., 2017), which is the range over which we see water column



modification. While iceberg melt itself is responsible for the freshening and cooling signal observed here, the sea ice matrix holding the icebergs in place in the proglacial fjord that supports this meltwater injection into the fjord over the course of several days. In this regard, it is the presence of a rigid ice mélange, rather than free floating icebergs, that leads to the observed water column stratification changes. The combination of hydrographic observations, plume modeling, and remote sensing of iceberg keel depths, all point to ice mélange melt being the primary driver for the observed stratification change above the depth of the outflowing subglacial discharge plume.

While we lack prior observational studies investigating the influence of ice mélange melt on the water column, several recent modeling studies have used general circulation models to address this unknown (Davison et al., 2020, 2022; Kajanto et al., 2023; Hager et al., 2023). In both Sermilik Fjord and Ilulissat Icefjord, where ice mélange remains present year-round, numerical simulations (Davison et al., 2020; Kajanto et al., 2023) show that the inclusion of icebergs invigorated the overall fjord circulation, in addition to causing significant cooling (~5°C in Sermilik Fjord and ~4°C in Ilulissat Icefjord) and freshening (0.7 PSU in both fjords) of surface waters. Both studies also found that the inclusion of icebergs was needed to reproduce the in situ hydrographic observations within the fjords. More recent idealized studies have found that for fjords where iceberg keel depths extend deeper than the entrance sill, the net cooling and freshening effect is larger with increasing iceberg concentrations within the ice mélange (Davison et al., 2022; Hager et al., 2023). Although the ice mélange event reported here is episodic and shows a smaller signal of water column modification than these modeling studies of persistent ice mélange, our observational results generally align. Furthermore, our observations provide constraints for future modeling efforts on understanding the influence of ice mélange melt on fjord stratification. The changes to water column stratification induced by this additional source of freshwater can have implications for the neutral buoyancy depth of subglacial discharge plumes, overall circulation in proglacial fjords, and the subsequent heat transport towards glacier termini.

### 4.2.1 Enhanced residence time of meltwater due to fjord circulation

While the addition of ice mélange meltwater is the primary contributor to the freshening and cooling signal observed in the proglacial fjord of Kangilliup Sermia, a recirculation gyre set up by subglacial discharge and surface water input is likely responsible for the prolonged signature of the short-term ice mélange event observed here. In situ water column velocity measurements show a cyclonic recirculation gyre to a depth of ~60 m in the fjord, with water flowing $0.142 \pm 0.063$ m s$^{-1}$ out of the fjord on the northern side, and at $0.154 \pm 0.049$ m s$^{-1}$ into the fjord on the southern side (Fig. 3c). This recirculation pattern can also be seen with optical imagery: as sediment-laden freshwater riverine plumes enter the fjord at the surface, they are deflected to the right (Fig. 1a; Supplementary Video). These in situ observations provide support for prior modeling studies of fjords of this type, which suggested recirculation patterns such as those observed could be explained by either internal Kelvin waves (Carroll et al., 2017) or standing eddies (Zhao et al., 2023). While we cannot distinguish the source of the observed recirculation pattern here with our limited snapshots of near-glacier water velocities, the mechanism that forms this





surface recirculation does not influence our findings that icebergs have an enhanced residence time in this fjord due to surface recirculations (Fig. S4). Together, our observations of water velocities along with previous fjord-modeling results (Carroll et al., 2017; Zhao et al., 2023) show that while these ephemeral ice mélange events are short-lived, the overall circulation pattern in fjords such as Kangilliup Sermia can cause icebergs to continuously recirculate near the glacier terminus. This recirculation
extends the overall time scale over which icebergs within the ice mélange can impart meltwater into the upper waters of the proglacial fjord.

## 4.3 Implications for fjords with summertime ice mélange

Previous studies across Greenland have noted that adjacent glaciers exhibit asynchronous dynamic behavior despite similar ocean forcings (Bartholomaus et al., 2016; Carroll et al., 2018; Fried et al., 2018; Catania et al., 2018; Carnahan et al., 2022;
King et al., 2018; Cowton et al., 2018; Fahrner et al., 2021). We argue that one missing discussion piece from these studies is the role that ice mélange melt may play in altering near-glacier ocean conditions, and how this influence varies between fjords. The observed freshening and cooling of the upper 200 m of water in front of Kangilliup Sermia illustrates that ice mélange presence can significantly alter water properties at least down to the depth of the outflowing discharge plume, and likely deeper to the depth of the iceberg keels (Fig. 3; Carroll et al., 2016; Slater et al., 2022). We assess the impact of different ice mélange
regimes on near-glacier hydrographic properties by comparing observations from Kangilliup Sermia to those of the neighboring Kangerlussup Sermia, as well as the ambient ocean water beyond the sill in *T-S* space (Fig. 4). While both fjords have persistent wintertime ice mélange, summertime ice mélange is only present at Kangilliup Sermia, allowing us to investigate the influence that the addition of ice mélange meltwater has on the surface water column stratification in two fjords that share the same ambient water source.


Water column properties in Kangerlussuup Sermia match those offshore to a depth of ~60 m, where the fjord water properties are slightly cooled, indicating the presence of outflowing glacially modified water at the fjord surface (Fig. 4; Carroll et al., 2016, 2018; Jackson et al., 2017). The water column properties in front of Kangilliup Sermia, however, deviate from the ambient ocean water at ~280-320 m depth due to the outflowing subglacial discharge plume (Figs. 3c and 4c; Carroll et al.,
2016; Slater et al., 2022). Above this depth, we see the water column properties follow the meltwater mixing line once again, indicating the presence of iceberg melt. This additional iceberg melt results in surface waters in the proglacial fjord of Kangilliup Sermia that are cooler and fresher than both the waters outside of the sill and the neighboring fjord, Kangerlussuup Sermia. Thus, at Kangilliup Sermia, the injection of cold, fresh water beneath the depth of the sill has the effect of progressively cooling the upper layer of the fjord over the course of the summer, aligning with previous modeling results (Davison et al.,
2022; Hager et al., 2023).

To quantify the impact that multiple ephemeral ice mélange events may have on fjord hydrography, we determine the number of events needed to explain the change in *S* between the near-glacier waters and those offshore. Using Eq. (4), we find that the



volume of meltwater needed to freshen the upper 200 m of the water column compared to the offshore water properties ($S_{offshore}$
$- S_{fjord}$ = 0.09 g kg$^{-1}$) is 1.42 x 10$^8$ m$^3$. Using the calculated meltwater volume from the ice mélange event discussed above
(5.47 x 10$^7$ m$^3$, described in Sect. 2.3), we estimate that the equivalent of ~2.6 events similar in magnitude to the one on August
5, 2014, are needed to match the observed differences in $S$. Manual inspection of cloud-free satellite imagery from Landsat 8
and Sentinel 2 during the summer of 2014 shows that ephemeral ice mélange events occurred at least ~4 times over the course
of the melt season, suggesting that the meltwater input from these ice mélange events is a reasonable explanation for how the
upper layer fjord waters are modified compared to offshore. The implication is that even short-lived ice mélange events have
the potential to cause lasting changes to water column stratification.

By instigating cooling throughout the entire water column for prolonged periods of time, the addition of ice mélange meltwater
to the proglacial fjord has the potential to reduce submarine melt rates at the glacier terminus in the summer months when
iceberg keel depth exceeds the depth of the sill (Davison et al., 2022; Hager et al., 2023). Overall, understanding how submarine
melting of icebergs can alter fjord circulation and subsequent heat exchange with the shelf is an important piece for modeling
the sensitivity of marine terminating glaciers to near-glacier fjord conditions, particularly for glaciers with ice mélange
conditions similar to those at Kangilliup Sermia. Prior work understanding ice mélange-ocean feedbacks has primarily focused
on tidewater glaciers with permanent ice mélange in their proglacial fjord, like Sermeq Kujalleq and Helheim Glacier. Our
observations show that even short-lived ice mélange events have the potential to cause lasting changes on water column
stratification in glacier-adjacent waters, suggesting that more work should be done to understand the dynamics and feedbacks
of ice mélange in systems with seasonal and ephemeral ice mélange presence. Ultimately, more hydrographic measurements
bookending ice mélange events are needed to constrain the exact extent to which meltwater alters fjord stratification and the
duration for which this altered stratification persists.



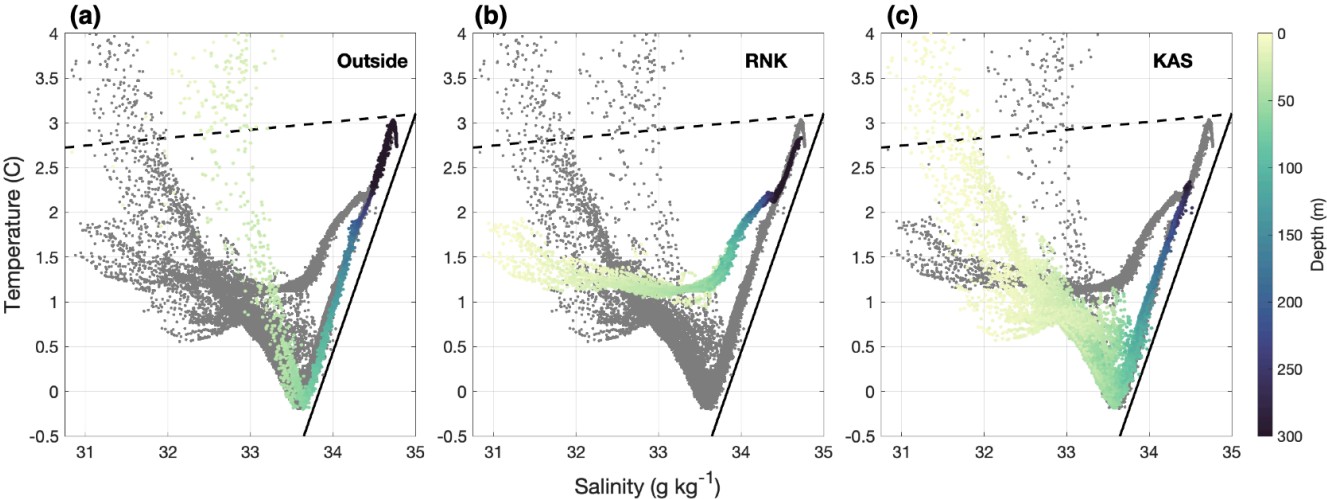


**Figure 4: Full-depth water column properties outside of the sill (a) and near the termini of Kangilliup Sermia (b) and Kangerlussuup Sermia (c) in *T-S* space, with water column properties colored by depth in the water column. Mixing lines with freshwater runoff (dashed line) and submarine meltwater (solid line) are superimposed.**

## 5 Conclusions

Through examining the first direct observations of hydrographic change induced by an ephemeral ice mélange event, we have shown that the meltwater released from even a short-lived ice mélange can alter the water column stratification by releasing fresh and cool water at depth in the fjord. In particular, we show that the integrated effect of several ephemeral ice mélange events over the summer melt season can explain the observed differences in hydrographic properties between the offshore ocean water and waters glacier-ward of the fjord mouth. We discount other mechanisms that could explain the observed water

column changes, such as advection and mixing of the water column, as well as enhanced freshwater input from non-iceberg sources. This enhanced meltwater input can particularly influence ice-ocean systems with deep grounding lines and shallow sills, where the subglacial discharge plume exits the fjord at a depth beneath the extent of ice mélange and the iceberg melt-modified waters can be recirculated. In addition, glaciers with deeper grounding lines, like Kangilliup Sermia, are often close to flotation and calve via buoyant flexure, which can lead to large episodic calving events that are more likely to produce

ephemeral ice mélange and the oceanographic changes observed here. On the other hand, fjords with shallow grounding lines, where the subglacial discharge plume exits the fjord at the surface and iceberg calving events tend to be smaller and more frequent, are less likely to be influenced by ice mélange melt either by its absence overall or by the fast export of any ice mélange meltwater with the outflowing plume. This work stresses the need for more high-temporal resolution coincident observations of both ice mélange characteristics (i.e., keel depth and concentration of icebergs within the sea ice matrix) and

fjord hydrography to fully capture the processes by which ice mélange meltwater can contribute to heterogeneous fjord characteristics around the Greenland Ice Sheet.



*Data availability.* Observational data from subsurface moorings have been previously archived at https://www.ncei.noaa.gov/archive/accession/0173969 (Catania et al., 2018). Additional hydrographic data from the CTD and
ADCP are being added to this link.

*Video supplement.* The time-lapse imagery of the terminus and proglacial fjord of Kangilliup Sermia from August 2014 will be available at the AV Portal of TIB Hannover upon publication.

*Supplement link.* The supplement related to this article is available on-line at *[LINK]*

*Author Contribution.* NA, DAS, and RP conceived the study. NA conducted the analysis and wrote the original manuscript. DAS, RP, GC, ELS, and TCB supported the interpretation of the results and contributed to the preparation of the manuscript. DAS, GC, JDN, ELS, LAS, and TCB developed the original ideas to conduct fieldwork and obtained the observations
presented here.

*Competing Interests.* The authors declare that they have no conflict of interest.

*Acknowledgements.* Funding was provided by NASA grant NNX12AP50G, NASA FINESST grant 80NSSC21K1594 to the
corresponding author, and partial support from the University of Oregon. We thank the captain and crew of the R/V *Sanna* for their contribution to field data collection.



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
