# Peer review of "Ice mélange melt changes observed water column stratification at a tidewater glacier in Greenland"

_EGUsphere, 2024_

## Referee Comment (RC2)

**General comments**

The authors have taken advantage of repeat oceanographic measurements in the Kangilliup Sermia fjord region to examine the influence of glacial mélange meltwater on water column temperature and salinity. Along with discussing the event and its impact within Kangilliup Sermia, the authors compare with Kangerlussuup Sermia and consider possible alternative processes that could influence observed changes. Overall, the paper is nicely organized, provides complete analysis of the observations, and presents a variety of useful visuals in the primary manuscript and supplementary materials. The results will be of interest and use for researchers examining ice-ocean interaction, modeling glaciated fjord environments, and considering system connections from glacier/ice sheet to ocean properties and onto biogeochemical processes.

For all figures the authors should check compatibility with colorblind requirements. They might also consider introducing different symbol types when appropriate to help to distinguish datasets.

Finally, I've included a variety of mostly minor comments below. After completing my initial review, I also read through the comments from Benjamin Davison and overwhelmingly agree.

**Specific comments (by line number)**

1. Consider shortening title to "Ice mélange melt changes observed water column stratification at Greenland tidewater glacier"

12. "brash ice" is only used twice in the manuscript – suggest using an alternative in both places and avoiding the phrase

30. correct to "marine-terminating"

31. Rather than use "these glaciers", suggest specifying the glaciers in question again. E.g., "The rapid retreat of Greenland marine-terminating glaciers…". It is easy for use of "these/this" type of words to be confusing and I recommend checking this across the manuscript. I noted similar instances of confusion here: "where" in line 351, "This" in line 396.

32-34. The note in this sentence (and in the last sentence in the paragraph) feels out of place to me or perhaps a not-so-useful transition. The topic within this sentence is rigid mélange influence on ice dynamics/retreat. But this paper really focuses on mélange melt. This could link with other ice-ocean processes that influence ice dynamics/retreat (noted line 49), but I find the focus on mélange rigidity re: glacier dynamics perhaps unnecessary.

It does strike me that this connects with the comments from Benjamin Davison re: ~line 300. The authors might think more about how much or little to discuss mélange rigidity across the paper and edit accordingly.

45. remove "of"

49-50. The sentence is easier to read and shorten when writing "increasing glacier and ice mélange submarine melting". Consider if similar changes can help in other parts of the manuscript. (This is one of many excellent writing tips from the recommended Writing Science book by Joshua Schimel.)

69, 73, 76. Recommend adding information on the bathymetric uncertainties in this region. Those vary widely across Greenland and would be helpful context for the reader. Similarly, including information on maximum fjord depth in this paragraph.

75. remove "-1,000 m;"

86. It would be useful in this paragraph to introduce a clear definition/distinction between icebergs and ice mélange that can be used throughout the manuscript. This paragraph would also benefit from including mention of the time periods evaluated by Sulak et al. (2017) and any note on whether there's an expectation of substantial change between that observation period and the one used within this paper's research.

87. remove "similarly"

Figure 1. (a) would benefit from slightly more satellite image viewable on the right and could be balanced by a small reduction on image left. It would also be useful to have the sill locations indicated in (a) and consider adding the tracks from (c) into the map-view in (a) (they could even have hash marks to help viewers align the data in (c)). In the caption, it would be helpful to add the rough time period for clearing at the end of the sentence noting formation over 6 hours (e.g., x hours or z days).

106. change to "event discussed here"

135-136. Why 250 m plume width? Also, why use the post-ice mélange CTD casts for the initial plume model stratification instead of pre-ice mélange? Can the authors also provide a note on uncertainty related to the plume model and what that implies for confidence on neutral buoyancy depth?

163-166. The second sentence here is confusing re: varying the depth range – please edit for clarity.

167. The authors note here that runoff/subglacial discharge don't vary substantially. Looking at Figure S7, an initial read would suggest a notable reduction in runoff (~300 to 100 m3/s) during the mélange event when only looking at the runoff record. Providing

comparative numbers (runoff vs mélange melt) or an alternative justification (e.g., line 284 about runoff vs subglacial discharge) could be helpful to convince the reader of the reasonableness of this approach.

180+. The authors note that the ice mélange broke up, moved down-fjord, and most of the ice mélange was transported out of the fjord. How much do you expect that freshwater is going to circulate and transport out along with mélange? In other words, what might you speculate about freshwater changes between August 9 (fjord cleared of ice) and August 11 (date of observations)?

Figure 2. I don't understand why the Aug 11 ~0-20 m water column is warmer than the Aug 4 0-20 m temperatures and this isn't explained in the text. Perhaps some of the information in the sentence at lines 206-208 is meant to help (noting freshwater surface input), but I was no less confused after reading this sentence. Or is the note at line 286-287 meant to address this?

Figure 3. In (a) it would be helpful to label some of the depth squares. At first the data I expected in (b) based on (a) and the caption note didn't seem to line up with (b) until I realized that I wasn't identifying the squares in (a) properly. They are very hard to see and it can be difficult to tell black from blue, so some help there would be nice. Please double check all the along-transect plots re: color/direction. For (c), it says that toward-glacier flow is positive (red) and that distance along fjord begins in the south at 0. These appear to mismatch – the top right corner of (c) should be southside toward-glacier flow, not northside toward-glacier flow (and based on line 217 sentence). It does get confusing since the toward-glacier flow pattern is reversed between the surface and below 100 m. Consider if further editing can help keep this clear for the reader.

231. Clarify "highest at the ocean surface" (not subaerial)

279. Note the location/boundaries of the warm temperature anomaly

298-300. This sentence does not read correctly – please rewrite.

300-301. Suggest using "facilitates" instead of "leads to"
Note: I agree with Benjamin Davison's comments on this topic and that modifications to this explanation are warranted.

394. Suggest specifying "Kangilliup Sermia fjord"

Figure S4. Add information to understand north/south on these plots.

Figure S9. What are the black triangles in (a)?

Table S1. Suggest stating the "full water column" depth range in the caption.

---

## Author Comment (AC1)

Summary: the authors present new hydrographic observations in two of Greenland's fjords that quantify the substantial impact of a 4-day period of ice mélange melt on water column properties. The observations are compared to output from standalone plume and iceberg-melt models to demonstrate that the observed changes in water column properties can be explained by ice mélange melt during the period of mélange presence, in addition to contributions from a subglacial discharge-driven plume.

Overall, the paper is very well-written, clear and convincing. It provides much needed observational evidence that support and add more detail to several modelling studies of iceberg-ocean interaction, which I think will be a great interest to the community.

I have just a few minor suggestions, questions and comments listed below, but otherwise I think the study is ready for publication:

The authors thank Dr. Davison for reading our manuscript and providing a thorough and positive review, the suggestions in which will be beneficial to our paper. We have provided responses to Dr. Davison's comments and will modify our manuscript accordingly in the next submission.

Minor comments

Line 25: can you go any further with this statement? Do your observations support the notion that ice mélange meltwater needs to be included (or its effects parameterised) in ocean circulation models or in ocean boundary conditions used to force glacier models? As written, this is implied but it's not clear the absence of a direct statement is deliberate.

Our observations do support the fact that ice melange meltwater needs to be included in both ocean circulation models and as forcing for tidewater glacier models for glaciers with deep grounding lines and high ice fluxes. To make this more clear, this sentence will be rewritten as "These observations provide critical constraints for and agreement with recent modeling studies that have suggested ice melange meltwater needs to be included in ocean circulation models for glaciers with deep grounding lines and high ice fluxes, which are precisely the glaciers exhibiting the largest magnitude terminus retreats at present".

Line 29: I'm not sure that "ice tongues" is the correct terminology here? I think ice tongues strictly refers to small, ice shelf-like features, which there are a few of in Greenland, but I don't think they existed at the majority of its now marine-terminating glaciers as the second part of the sentence implies.

While the breakup of ice tongues and replacement with ice melange has been documented at large glaciers like Sermeq Kujalleq, we agree that this is not the best terminology for our focus on marine-terminating glaciers that did not historically have ice tongues. Therefore, we

will remove the focus on ice tongues from this opening sentence and  will combine it with the next to read "Ongoing observations have documented the rapid retreat and dynamic thinning of Greenland's marine-terminating outlet glaciers (e.g., Greene et al., 2024; King et al., 2020), which has been attributed to environmental forcings occurring at the ice-ocean boundary (Carnahan et al., 2022; Murray et al., 2010; Nick et al., 2009)."

Line 34: consider "days to years", since this study focuses on a 4-day period of mélange.

This will be changed in the manuscript.

Line 36: "the influence of ice mélange meltwater and its temporal variability" seems a bit ambiguous to me and somewhat contradictory to the first part of the sentence that cites several modelling studies that examined the influence of ice mélange meltwater on fjord water properties and circulation. Can you rephrase to make your meaning clearer, which I think is that the cited studies focused on long-lived mélange rather than ephemeral mélange?

This statement was intended to highlight that we lack observational data to confirm the modeling studies cited. We will rephrase this statement to "Although several numerical modelling studies have suggested that meltwater from ice melange can alter the ocean forcing near tidewater glaciers (Davison et al., 2020, 2022; Kajanto et al., 2023; Hager et al., 2023), this process is yet to be confirmed by observations".

Lines 46/47: Can you be clearer about the directionality of the impact of changes in freshwater flux on fjord-shelf exchange? i.e. does an increase in freshwater flux enhance or reduce exchange? As written, it states that any change in freshwater flux corresponds to an increase in exchange, which doesn't seem right to me.

This statement will be reworded from "Changes in the freshwater flux exiting these glacial fjords can enhance exchange…" to "Increases in the freshwater flux exiting these glacial fjords can enhance exchange" in order to improve clarity when describing this feedback.

Equation 6: Shouldn't $A_{ice}$ be the total iceberg-ocean contact area, not the area of the fjord surface covered by ice mélange?

Ideally, Aice should be the total iceberg-ocean contact area rather than the surface area of the fjord covered by ice melange. As we do not have direct measurements of this underwater surface area, we use the surface area of the fjord as a lower bound for this calculation. The actual iceberg-ocean contact area will be larger than the surface area of ice melange cover, which would lead to a reduction in the total length of time needed to melt the volume of ice melange predicted. For example, Sulak et al. (2017, *Annals of Glaciology*) used remote sensing to investigate icebergs in the proglacial fjord of Kangilliup Sermia

during a similar time examined here (summers of 2013-2015). They found the average cross-sectional areas of icebergs in the fjord and calculated what their keel depths would be if the iceberg maintained its shape throughout its depth (i.e., was a block shape). They found that the average submerged surface area of icebergs in this fjord was 41.34 km2. The surface area that we calculate for Aice by outlining satellite imagery of the ephemeral ice melange was 29.56 km2. Therefore, using the estimate from Sulak et al. (2017) we can infer that the actual iceberg-ocean contact area in our ice melange is roughly ~1.4x higher than the value for Aice that we have used in this manuscript. This would reduce the duration of time required to produce enough ice melange meltwater to realistically cause the observed changes in salinity observed, and would further strengthen our findings. To clarify this, we will add a statement to this effect in the new version of the manuscript.

Lines 191-194: can you clarify what the errors on the T and S measurements represent? As presented, they seem large relative to the observed changes in T and S, which might or might not be important for the interpretation depending on how the errors are defined.

The reported numbers for T and S are the transect-averaged measurement +/- one standard deviation from this average. To obtain these, we take the average of all CTD casts in the pre- and post-ice melange transects respectively. We then average these casts over the depth range in question (full water column and 200 m). The reported ranges are obtained by taking the standard deviation over the same depth range which the casts are averaged over, so they show the spread of the data within that depth region and not an error. We will specify this in the new version of the manuscript.

Line 203: erroneous "then"?

This will be removed from the manuscript.

Line 229-232: are these melt rates from icebergs with a particular keel depth? Or averaged across icebergs with a range of keel depths? I think this should be specified in the text.

These melt rates are averaged over all modeled iceberg keel depths. This will be specified in the text by changing "Modeled iceberg melt rates vary between 0.08-1.40 m/d (Fig. S3)..." to "The total range of iceberg melt rates is between 0.08-1.40 m/d for all modeled keel depths…"

Line 242: "these time spans for complete melting of the ice mélange" – I found this a bit confusing. Wasn't the preceding paragraph providing time-spans for melting of 14% of the ice mélange? So it is expected that 86% of the mélange must have been exported as solid ice? (plus or minus the uncertainties relating to the initial ice volume, but the main thing here is that there was more than enough ice available). I recommend providing a figure to illustrate the time-spans required to produce Vmelt under the range of estimated melt rates.

We will clarify this in the new version of the manuscript. We agree that the wording currently used makes it sound like the time estimates stated are for 100% of the ice melange to melt, when instead it should read that these time estimates are for the 14% of icebergs that needed to melt to explain the observed changes in salinity. We will add a figure to the supplementary information showing how the time-span to produce Vmelt varies with the different melt rate estimations.

Line 252: erroneous "surface"?

This will be removed from the manuscript.

Line 300 and surrounding text: I'm not sure that evidence has been presented to support this statement. Hydrographic observations are only available before and after the icebergs were present in the fjord, so there is no information about how the water properties changed during that period of 4 days – the modification may have happened steadily over the 4-days or it may have mostly happened in the first day. We do know that the icebergs were only held in place for ~12 hours before becoming mobile, which either suggests that rigid mélange caused all of the modification in the first 12 hours, or that modification continued despite the mélange become mobile.

This statement was intended to highlight the difference between the impact of ice melange on water column properties as opposed to free floating icebergs from a typical calving event. We recognize that we do not have evidence identifying the exact time span over which the water column modification occurred over the course of the 4 days, but we believe that the formation of a briefly rigid ice melange increased the residence time over which the calved icebergs were able to inject meltwater into the glacier-adjacent water column.

To make this more clear, we will address the limitations of this data by rephrasing the statement to "While iceberg melt itself is responsible for the freshening and cooling signal observed here, the sea ice matrix holding the icebergs in place in the proglacial fjord supports this meltwater injection into the fjord by increasing the residence time of icebergs in the glacier-adjacent water column. In this regard, it is the presence of ephemeral ice mélange, rather than free floating icebergs, that facilitates the observed water column stratification changes."

Line 358/9: "progressively cooling the upper layer of the fjord over the course of the summer" – I think it would be clearer if this statement came after the explanation given the following paragraph. I also think it would be relevant here to state whether any ephemeral mélange events occurred before the one described here, and if so give the dates of that occurrence and describe how that might or might not have affected the pre-event water properties.

We believe that this statement helps introduce the calculation done in the next paragraph and connect the results in the current paragraph to numerical modeling results done in prior studies. Therefore, we would like to leave this sentence at its current location.

It is difficult to quantify the exact number of ephemeral ice melange events that occur in the proglacial fjord of Kangilliup Sermia due to the temporal resolution of satellite imagery and the fact that ice melange can remain frozen in place as short as 12 hours like the example discussed here. We state on line 368 that ephemeral ice melange occurs at least 4 times during the summer of 2014, but we agree that adding the estimated dates of these events would be helpful for interpretation and they will be added to the next version of the manuscript.

The prior ephemeral ice melange events that occurred in the fjord during the summer of 2014 would have progressively cooled and freshened the upper layers of the water column. We can see the effect of this in our pre-ephemeral ice melange CTD cast on August 4 in Figures 2 and 4, where the surface waters in the fjord of Kangilliup Sermia are cooler and fresher than the water column at the 'Outside' mooring. The change in our hydrographic observations between August 4 and August 11 help isolate the influence of a single one of these ephemeral ice melange events, and it is likely that other events in the fjord would create similar water column changes (with the exact magnitude dependent on the size and duration of the ice melange in question). We will add a statement explicitly acknowledging this in the next version of the manuscript.

Paragraph starting line 361: I think this analysis assumes that the mélange events have no impact on 'offshore' water properties. The offshore properties used here are from a co-located CTD-cast and mooring outside of the sill where the two fjords branch, some 60 km from the glacier terminus. If the authors agree, I think this assumption should be acknowledged and the direction of impact on this analysis should be given.

The authors agree and will include this assumption in the new version of the manuscript as:

"As the offshore water properties are taken from a mooring located ~60 km from the glacier terminus, this calculation assumes that meltwater from ice mélange has not altered the offshore water properties. If ice mélange meltwater did in fact modify the offshore water properties, a smaller meltwater volume would be needed to explain the difference in salinity between the two locations, indicating that fewer ephemeral ice mélange events would be needed to match the observed differences in S."

I think it would be worth adding a paragraph to the discussion examining why ephemeral mélange events occur at some fjords but not others (some combination of mid-depth plume outflow and large calving events, but not so much calving that you get near-permanent mélange?), or at least to give some indication as to how widespread they are or might be

around Greenland. As written, some readers could conclude that most fjords are either like Kangerlussuup Sermia without ephemeral mélange events, or they are like Sermilik Fjord where there is near-permanent mélange, and only a few have these ephemeral events.

There currently are no published Greenland-wide inventories of the fjords in which ephemeral ice melange forms. A master's thesis by Emma Swanninger (2020, University of Idaho) looked at a set of 24 glaciers around the margin of Greenland between 2000-2019 and found that 11 of those experienced ephemeral ice melange in the summertime. We believe that investigating what causes ephemeral ice melange events is beyond the scope of this study, as the hydrographic changes indicated here for ephemeral events are also likely observed at systems with permanent ice melange. However, we will add a reference to Swanninger's thesis to show that ephemeral ice melange occurs at quite a few fjords around Greendland's margins.

Figure 3a: I think it would help if labels I, II and III were briefly summarised in the figure caption, as well as in the main text, just to save the reader scrolling around the manuscript.

This will be added to the new version of the manuscript.

Figure 3a: I found the square markers a bit confusing. The caption states they are at intervals of 50 m, but I count 9 black square markers before (above?) the panel (b) inset, which includes data from 50-100 m. Does that just mean the water column properties don't change smoothly in T-S space with depth? If so, then I'm not sure that the square markers aid the interpretation of panel (a). Or should I just be comparing the position of a black square with a position of a blue square in T-S space? If so, then is there some way you can make it clear which squares correspond to the same depth on the black and blue curves?

The square markers are intended to allow the reader to determine relative position in the water column along the data profile. Deep in the water column, the temperature and salinity are largely uniform and so many of the markers in the deepest section of the profile (top right) are stacked on top of eachother. While we present data for the entire water column in this manuscript, the changes induced by the ice melange and subglacial discharge plume occur in the upper ~400 m of the water column, which have markers visible in the plot. To aid in interpretation, we will add labels to the plot every 100 m in black and blue text.

Figure 4: what do the grey dots represent? I can't see a description of them in the caption. Please can you also specify the dates of the presented CTD casts? It's not clear if the RNK data is from before or after the mélange event.

Each panel of the figure shows all data from 'Outside', 'RNK', and 'KAS'. To facilitate comparison between the region being emphasized and the other regions' water properties, we highlight the region of focus in each panel with the blue-green colorbar and leave the

other regions' properties as grey dots. We will specify this in the figure caption in the updated version of the manuscript.

The hydrographic data presented in this figure comes from all CTD casts collected during the field campaign in the summer of 2014, which took place between July 29 and August 12. We will add a table to the supplemental information stating the dates of the CTD casts presented in this figure.

Benjamin Davison

---

## Author Comment (AC2)

General comments

The authors have taken advantage of repeat oceanographic measurements in the Kangilliup Sermia fjord region to examine the influence of glacial mélange meltwater on water column temperature and salinity. Along with discussing the event and its impact within Kangilliup Sermia, the authors compare with Kangerlussuup Sermia and consider possible alternative processes that could influence observed changes. Overall, the paper is nicely organized, provides complete analysis of the observations, and presents a variety of useful visuals in the primary manuscript and supplementary materials. The results will be of interest and use for researchers examining ice-ocean interaction, modeling glaciated fjord environments, and considering system connections from glacier/ice sheet to ocean properties and onto biogeochemical processes.

For all figures the authors should check compatibility with colorblind requirements. They might also consider introducing different symbol types when appropriate to help to distinguish datasets.

Finally, I've included a variety of mostly minor comments below. After completing my initial review, I also read through the comments from Benjamin Davison and overwhelmingly agree.

The authors thank Dr. Moon for reading our manuscript and providing a thorough and positive review, the suggestions in which will be beneficial to our paper. We have provided responses to Dr. Moon's comments and will modify our manuscript accordingly in the next submission.

Specific comments (by line number)

1. Consider shortening title to "Ice mélange melt changes observed water column stratification at Greenland tidewater glacier"

The authors will shorten the title to "Ice melange melt changes observed water column stratification at a tidewater glacier in Greenland". This will shorten the title, but also keep the focus of it on tidewater glaciers rather than Greenland specifically.

12. "brash ice" is only used twice in the manuscript – suggest using an alternative in both places and avoiding the phrase

The phrase brash ice will be removed from the manuscript in the abstract and on line 12.

30. correct to "marine-terminating"

This will be corrected in the new version of the manuscript.

31. Rather than use "these glaciers", suggest specifying the glaciers in question again. E.g., "The rapid retreat of Greenland marine-terminating glaciers...". It is easy for use of

"these/this" type of words to be confusing and I recommend checking this across the manuscript. I noted similar instances of confusion here: "where" in line 351, "This" in line 396.

Vague language such as the examples mentioned here will be changed accordingly throughout the manuscript.

32-34. The note in this sentence (and in the last sentence in the paragraph) feels out of place to me or perhaps a not-so-useful transition. The topic within this sentence is rigid mélange influence on ice dynamics/retreat. But this paper really focuses on mélange melt. This could link with other ice-ocean processes that influence ice dynamics/retreat (noted line 49), but I find the focus on mélange rigidity re: glacier dynamics perhaps unnecessary.

It does strike me that this connects with the comments from Benjamin Davison re: ~line 300. The authors might think more about how much or little to discuss mélange rigidity across the paper and edit accordingly.

We believe that some mention of rigidity is important here, as it highlights the difference between the impact of ice melange on water column properties as opposed to free floating icebergs from a typical calving event. The rigidity of the ice melange, even if only for ~12 hours, increases the residence time of icebergs in the glacier-adjacent water column. The authors plan to rephrase their focus on rigidity to remove the discussion of its impact on glacier dynamics, as that is likely minimal with the short-lived duration of the ice melange presented here. Instead, the authors will refocus the discussion of rigidity on how it prolongs the residence time of icebergs in the proglacial fjord.

45. remove "of"

This will be removed in the manuscript.

49-50. The sentence is easier to read and shorten when writing "increasing glacier and ice mélange submarine melting". Consider if similar changes can help in other parts of the manuscript. (This is one of many excellent writing tips from the recommended Writing Science book by Joshua Schimel.)

This specific instance will be changed in the manuscript and we will look through the rest of the manuscript for additional examples of this.

69, 73, 76. Recommend adding information on the bathymetric uncertainties in this region. Those vary widely across Greenland and would be helpful context for the reader. Similarly, including information on maximum fjord depth in this paragraph.

We will add a statement on bathymetric uncertainty to this paragraph, which is +/- 10 m in this region. In addition, we will add the maximum fjord depth for both RNK and KAS to this paragraph, which are 1,100 m and 620 m respectively.

75. remove "-1,000 m;"

This will be removed in the manuscript.

86. It would be useful in this paragraph to introduce a clear definition/distinction between icebergs and ice mélange that can be used throughout the manuscript. This paragraph would also benefit from including mention of the time periods evaluated by Sulak et al. (2017) and any note on whether there's an expectation of substantial change between that observation period and the one used within this paper's research.

We believe that our definition of ice melange in the introduction section is sufficient, as we specify that ice melange is the frozen conglomeration of icebergs and sea ice.

Sulak et al. (2017) evaluated icebergs in the fjord of Kangilliup Sermia and Kangerlussuup Sermia in the summers of 2013, 2014, and 2015. This bookends the observations presented here, so we do not expect any change in iceberg distributions between our research and the observations presented in Sulak et al. (2017).

87. remove "similarly"

This will be removed from the manuscript.

Figure 1. (a) would benefit from slightly more satellite image viewable on the right and could be balanced by a small reduction on image left. It would also be useful to have the sill locations indicated in (a) and consider adding the tracks from (c) into the map-view in (a) (they could even have hash marks to help viewers align the data in (c)). In the caption, it would be helpful to add the rough time period for clearing at the end of the sentence noting formation over 6 hours (e.g., x hours or z days).

Figure 1 will be modified as suggested by shifting the focus of the satellite image towards the ice sheet, annotating the sill locations, and adding the location of the bathymetry profile from (c) into panel (a). The time period for ice melange clearing will be added to the caption.

106. change to "event discussed here"

This will be changed in the manuscript.

135-136. Why 250 m plume width? Also, why use the post-ice mélange CTD casts for the initial plume model stratification instead of pre-ice mélange? Can the authors also provide a note on uncertainty related to the plume model and what that implies for confidence on neutral buoyancy depth?

A 250 m plume width is used as work done by Jackson et al. (2017) focused on Kangerlussuup Sermia found that a line plume of ~200 m width best matched observations in this region. We quantify the uncertainty of different plume widths by varying the plume width between 100-500 m in our model (as discussed on Line 135) and presenting these results in the Supplementary Information (Figure S8). Despite the

large range of tested plume widths, the subglacial discharge plume always reaches neutral buoyancy at depth in the fjord (between 100 - 250 m deep).

We use the post-ice melange CTD casts so that we can more easily compare our predicted plume neutral buoyancy depth with the water column velocity observations from the ADCP presented in the main text of the manuscript.

163-166. The second sentence here is confusing re: varying the depth range – please edit for clarity.

We will change this from "We vary the depth range considered between 100-300 m…" to "We vary the maximum depth considered between 100-300 m…" to clarify that we are only testing the sensitivity of our results to the size of the control volume.

167. The authors note here that runoff/subglacial discharge don't vary substantially. Looking at Figure S7, an initial read would suggest a notable reduction in runoff (~300 to 100 m3/s) during the mélange event when only looking at the runoff record. Providing comparative numbers (runoff vs mélange melt) or an alternative justification (e.g., line 284 about runoff vs subglacial discharge) could be helpful to convince the reader of the reasonableness of this approach.

We agree that there is a reduction in runoff during the ephemeral ice melange event investigated here, although the authors still believe it is negligible in the calculation of ice melange meltwater volume. As the rate of subglacial discharge decreases between the two hydrographic cast dates, the contribution of subglacial discharge to the control volume would be less in the second water column profile than the first. This would only increase the influence of the ice melange melt on the water column. We will highlight this impact in the new version of the manuscript by noting that the runoff does not substantially increase in between our two data collection dates.

180+. The authors note that the ice mélange broke up, moved down-fjord, and most of the ice mélange was transported out of the fjord. How much do you expect that freshwater is going to circulate and transport out along with mélange? In other words, what might you speculate about freshwater changes between August 9 (fjord cleared of ice) and August 11 (date of observations)?

This is difficult to quantify without observations during this time period, but we do expect that some of the freshwater from the ice melange event was likely transported out of the fjord during this gap in observations. We expect that the icebergs were likely exported from the fjord before the meltwater was, as icebergs are subject to both the variable ocean currents in the submarine environment, as well as strong katabatic winds that can clear the smaller icebergs in the fjord. In addition, numerical modeling results of fjords with geometries similar to Kangilliup Sermia have shown that ice melange meltwater is typically recirculated into the fjord at the sill, leading to a longer residence time of the meltwater in the fjord than the icebergs (Davison et al., 2022; Hager et al., 2024, The Cryosphere). While we don't have direct observations of the water column during this time period, we would expect that the magnitude of cooling and freshening

observed here would have been even larger had they been taken directly after the ice melange breakup. A logical next step to further the research presented in this paper would be to do a focused study on water column change in a fjord with ephemeral ice melange by completing CTD casts before, during, and after events. This would help establish the exact timeline over which the water column is transformed.

Figure 2. I don't understand why the Aug 11 ~0-20 m water column is warmer than the Aug 4 0-20 m temperatures and this isn't explained in the text. Perhaps some of the information in the sentence at lines 206-208 is meant to help (noting freshwater surface input), but I was no less confused after reading this sentence. Or is the note at line 286-287 meant to address this?

The notes in lines 206-208 and 286-287 are meant to help explain this, but the authors will make these points more explicit in the next version of the manuscript. While the upper 20 m are warmer after the ice melange event, we note that the water column remains much fresher in this layer after the event. Two main mechanisms could have contributed to this. First, as noted in the lines mentioned above, the waters at the surface of the water column are pulled back towards the runoff mixing line in the upper 20 m. This suggests that the iceberg melt-modified intermediate waters (50-100m) are being mixed with a surface input of freshwater, which is likely terrestrial runoff.

An additional mechanism that could have contributed to this warming is atmospheric heat flux. The cold and fresh meltwater layer is subject to heating from the atmosphere, which could have contributed to a warming of the upper 20 m of the water column. Unfortunately the glacier-adjacent weather station was recovered just before the ephemeral ice melange event investigated here, so we are unable to directly calculate the heat flux added to the water column during this time period. However, time-lapse video footage of the proglacial fjord shows that the weather was sunny for the duration of the ephemeral ice melange event.

We estimate that ~30 W/m2 of net heat flux would be needed to cause the observed warming in the upper 20 m of the water column using the equation $Q_{net} = (dT/dt)*rho*Cp*Z$, where $dT/dt$ is the change in temperature in the upper 20m of the water column between our two observations, rho is the average seawater density, Cp is the heat capacity of seawater, and Z is the depth being examined. ERA5 Daily Averaged reanalysis data shows that the estimated solar radiation between our two observations is ~48 W/m2, suggesting that it is possible atmospheric heat flux contributed to the surface warming of the water column observed here.

Figure 3. In (a) it would be helpful to label some of the depth squares. At first the data I expected in (b) based on (a) and the caption note didn't seem to line up with (b) until I realized that I wasn't identifying the squares in (a) properly. They are very hard to see and it can be difficult to tell black from blue, so some help there would be nice. Please double check all the along-transect plots re: color/direction. For (c), it says that toward-glacier flow is positive (red) and that distance along fjord begins in the south at 0. These appear to mismatch – the top right corner of (c) should be southside toward-glacier flow, not northside toward-glacier flow (and based on line 217 sentence).

It does get confusing since the toward-glacier flow pattern is reversed between the surface and below 100 m. Consider if further editing can help keep this clear for the reader.

Depth markers in panel (a) will be labeled every 100 m to facilitate easier interpretation of the figure.

The authors appreciate the note about the along-transect plots, as this highlighted an error in the figure caption. Distance does indeed increase from north to south, so the figure caption should state that "Distance across fjord increases southward" rather than "Distance across fjord increases from southward". This will be changed in the manuscript. The color scale remains correct, and with this change in the figure caption the recirculation gyre, where the ocean currents are away from the glacier in the north (i.e., negative velocities) and towards the glacier in the south (i.e., positive velocities) are shown in the blue and red currents respectively.

231. Clarify "highest at the ocean surface" (not subaerial)

This will be clarified in the manuscript.

279. Note the location/boundaries of the warm temperature anomaly

There is no specific warm temperature anomaly observed here. The authors are referring to work done by Carroll et al. (GRL, 2016), who showed that the subglacial discharge plume in deeply grounded systems, such as Kangilliup Sermia, will have higher temperature and salinity than the corresponding ambient ocean waters. This means that if the change in hydrography documented here was due to enhanced subglacial discharge, we would instead see an increase in temperature and salinity in our post-melange hydrographic profiles rather than the cooling and freshening observed.

298-300. This sentence does not read correctly – please rewrite.

This sentence will be rewritten to "While iceberg melt itself is responsible for the freshening and cooling signal observed here, the sea ice matrix holding the icebergs in place in the proglacial fjord supports this meltwater injection into the fjord by increasing the residence time of icebergs in the glacier-adjacent water column."

300-301. Suggest using "facilitates" instead of "leads to"
 Note: I agree with Benjamin Davison's comments on this topic and that modifications to this explanation are warranted.

The wording in the text will be modified from 'leads to" to "facilitates". To address Dr. Davison's concerns, the wording in this section will be changed to highlight that the limitation of our data at estimating the exact time period over which water column transformation occurs. However, we will still highlight that the formation of a briefly rigid ice melange increased the residence time over which the calved icebergs were able to inject meltwater into the glacier-adjacent water column, as this emphasizes the

difference between the results presented here and those surrounding free-floating icebergs.

394. Suggest specifying "Kangilliup Sermia fjord"

This will be specified in the new version of the manuscript.

Figure S4. Add information to understand north/south on these plots.

Similar to Figure 3 in the main manuscript, a statement indicating that "distance across fjord increases southward" will be added to the figure caption.

Figure S9. What are the black triangles in (a)?

The black triangles are the locations of the CTD casts used in this manuscript. This information will be added to the figure caption.

Table S1. Suggest stating the "full water column" depth range in the caption.

The full water column depth range (5-800 m) will be added to the table caption.